



# Crowdsourced Air Traffic Data from the OpenSky Network 2019–20

Martin Strohmeier[1,2,4], Xavier Olive[3,4], Jannis Lübbe[4], Matthias Schäfer[4,5], and Vincent Lenders[1,4]

[1]armasuisse Science + Technology
[2]University of Oxford
[3]ONERA DTIS, Université de Toulouse
[4]OpenSky Network
[5]University of Kaiserslautern

**Correspondence:** Martin Strohmeier (strohmeier@opensky-network.org)

**Abstract.** The OpenSky Network is a non-profit association that crowdsources the global collection of live air traffic control data broadcast by airplanes and makes it available to researchers.

OpenSky's data has been used by over a hundred academic groups in the past five years, with popular research applications ranging from improved weather forecasting to climate analysis. With the COVID-19 outbreak, the demand for live and historic aircraft flight data has surged further. Researchers around the world use air traffic data to comprehend the spread of the pandemic and analyze the effects of the global containment measures on economies, climate and other systems.

With this work, we present a comprehensive air traffic dataset, derived and enriched from the full OpenSky data and made publicly available for the first time (Olive et al. (2020), DOI: https://doi.org/10.5281/zenodo.3928564). It spans all flights seen by the network's more than 3000 members between 1 January 2019 and 1 July 2020. Overall, the archive includes 41,900,660 flights, from 160,737 aircraft, which were seen to frequent 13,934 airports in 127 countries.

## 1   Introduction

In this paper, we present a dataset of global flight movements derived from crowdsourced air traffic control data collected by the OpenSky Network (Schäfer et al. (2014)), which are widely used in many fields, including several areas pertaining to Earth System Sciences. With the spread of COVID-19, they are furthermore widely used in the understanding of the pandemic and its effects.

OpenSky flight data has regularly been used in analyzing environmental issues such as noise emissions (Tengzelius and Abom (2019)) or black carbon article emissions (Zhang et al. (2019)) to name but a few. In the wake of the pandemic, OpenSky has received a surge of more than 70 requests for air traffic data specifically related to COVID-19. The research behind these requests can be largely separated into two different areas, *epidemiological modelling* and understanding the *systemic impact* of the pandemic.

The first category, modeling of the possible spread of COVID-19, was of crucial interest early in the stages of the pandemic and will again gain importance to estimate travel safety in the future. The utility of flight data for this purpose was illustrated for



example in widely circulated studies such as Bogoch et al. (2020) but has been known to be useful in the context of pandemics for much longer (e.g., Mao et al. (2015)).

The second main category comprises the analysis of the socio-ecological impact of COVID-19 and measures implemented to fight it. It uses flights for example as an indicator of economic activity (at a given airport, region, or globally) as illustrated in Miller et al. (2020). Examples of such use of data provided by OpenSky can be found in Bank of England, Monetary Policy Committee (2020), International Monetary Fund (2020) or United Nations Department of Economic and Social Affairs (2020).

Flight data can further be used to understand the impact of the sudden drop in air traffic on many global systems. For example, 30   Lecocq et al. (2020) employed OpenSky data recently in order to analyze the impact of COVID-19 mitigation measures on high-frequency seismic noise and we received several requests relating to research specifically on the xte impact of COVID-19. This present dataset, available at https://doi.org/10.5281/zenodo.3928563, was created in order to make it easier for researchers to access air traffic data for their own systemic analyses.

## 2   Background

Crowdsourced research projects are a form of 'citizen science' whereby members of the public can join larger scientific efforts by contributing to smaller tasks. In the past, such efforts have taken many forms including attempting to detect extra-terrestrial signals (UC Berkley (2019)), or exploring protein folding for medical purposes (Pande (2019)). Typically, the projects form distributed computing networks with results being fed to a central server.

In a parallel development, software-defined radios (SDRs) have become readily available and affordable over the past decade. 40   SDR devices present a significant change to traditional radios, in that wireless technologies can be implemented as separate pieces of software and run on the same hardware. This has greatly reduced the barriers to entry, so many more users can now take part in wireless projects such as crowdsourced sensor networks with little cost. This development has given rise to several global crowdsourced flight tracking efforts, from commercial to enthusiast and research use.

The concept of flight tracking itself is based on several radar technologies. Traditionally, these were expensive and inaccurate 45   non-cooperative radars developed for military purposes. With the explosive growth of global civil aviation, however, more accurate cooperative radar technologies have been deployed to ensure safety and efficiency of the airspace.

For this dataset of flight movements, we use the data broadcast by aircraft with the modern Automatic Dependent Surveillance – Broadcast (ADS-B) protocol. This data includes position, velocity, identification and flight status information broadcast up to twice a second (see Schäfer et al. (2014)). The protocol is being made mandatory in many airspaces as of 2020, resulting 50   in broad equipage among larger aircraft from industrialized countries and emerging economies as described by Schäfer et al. (2016).

Figure 2 illustrates the principle of OpenSky in the abstract: The data is broadcast by ADS-B-equipped aircraft and received by crowdsourced receivers on the ground, which have typical ranges of 100-500 km in a line of sight environment. The data is then sent to the OpenSky Network, where it is processed and stored in a Cloudera Impala database. In line with its mission as

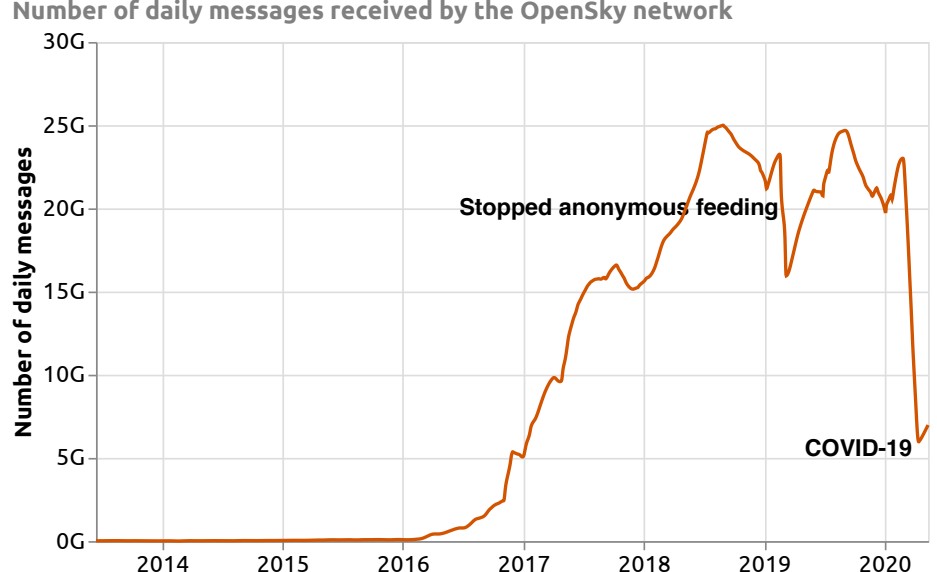

**Figure 1.** OpenSky message growth 2014-2020.

a non-profit organisation, OpenSky then grants researchers from academic and other institutions direct access to this database on request (Schäfer et al. (2014)).

Along with the global sensor coverage, the database has initially grown exponentially since its inception on 2014 (see Fig. 1) and currently comprises over 23 trillion messages, taking up around 2 Petabytes. In peak pre-pandemic times, almost 100,000 flights were tracked per day. The raw data available in the Impala database has been used in more than 100 academic publications as of July 2020. However, despite available application programming interfaces and third-party tools, the access to this data requires significant investment of time and resources to understand the availability and underlying structure of the database. With this data set and its accompanying descriptor we want to address this accessibility issue and make a relevant part of the OpenSky Network flight meta data accessible to all researchers.

## 3 Methods

### 3.1 Crowdsourced Collection

The raw data used to generate the dataset was recorded more than 3000 crowdsourced sensors of the OpenSky Network. The network records the payloads of all 1090 MHz secondary surveillance radar downlink transmissions of aircraft along with the *timestamps* and *signal strength indicators* provided by each sensor on signal reception. Part of this data collection are the exact aircraft locations broadcast at 2 Hz by transponders using the ADS-B technology.



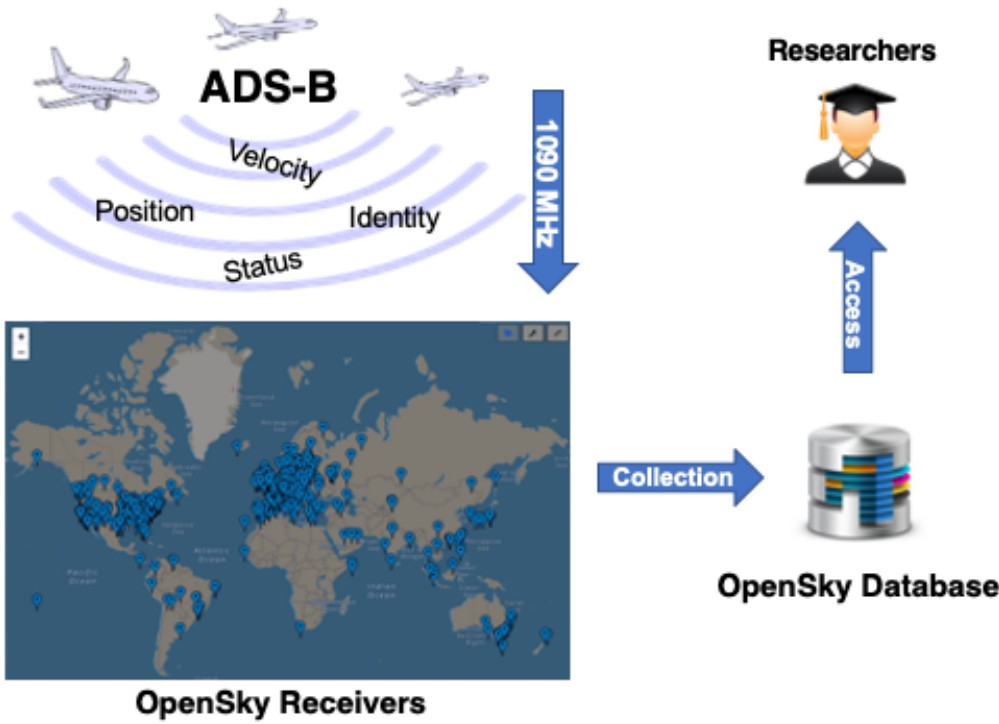

**Figure 2.** High-level illustration of the flight data crowdsourcing process, including map of active receivers on July 1, 2020. © OpenStreetMap contributors 2020. Distributed under a Creative Commons BY-SA License.

As the data comes from a crowdsourced system of receivers, it is dealing with numerous challenges and difficulties found in such an organically-grown, non-controlled set of receivers. However, it is the only feasible option for the large-scale collection of open research data as collecting data from a synchronized and controlled deployment would be less flexible and less widely applicable, in particular for a non-profit research endeavour. Conversely, due to the high sensor density and high level of redundancy in the OpenSky Network, many well-covered regions of this data achieve the quality of controlled deployments on

a nation-wide level in many countries.

     The true coverage of the network, i.e. actually received positions of airplanes, is illustrated in Fig. 3, both for 1 January 2019 and during the pandemic on 1 May 2020. Historic coverage for any given day is visible on https://opensky-network.org/network/facts.

## 3.2    Derivation of Flights

We define a *flight* for the purpose of this dataset as the time between the first received ADS-B contact of one specific aircraft and the last. A flight must be of at least 15 minutes. If a flight leaves OpenSky's coverage range for more than 10 minutes, it is principally considered finished at the point of last contact. To prevent counting flights multiple times if they return into the coverage range after more than 10 minutes (e.g., for any flight over the Atlantic Ocean), we apply a simple check: if time,



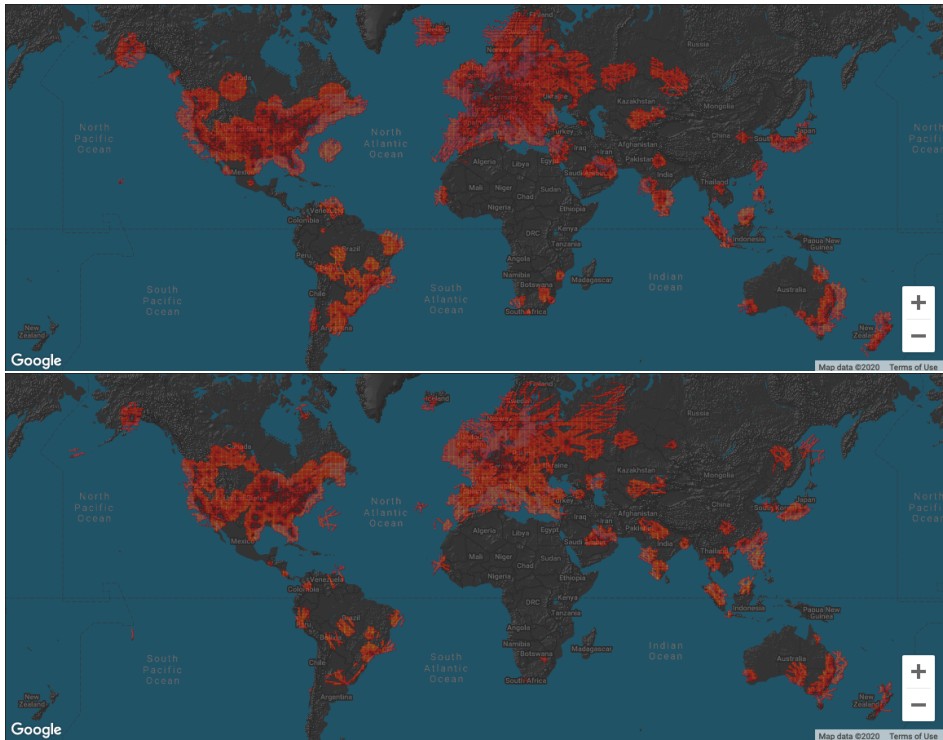

**Figure 3.** Coverage of OpenSky on 1 January 2019 and 1 May 2020. © Google Maps

distance and reported velocity match, they will be considered segments of the same flight. If not, it is assumed that the aircraft

has landed at some point.

The destination airport candidates are received from these identified flight trajectories as follows. If the last position seen is above 2500 meters, no candidate is defined and the value is set to 'NULL'. Else, the descending trajectory is extrapolated towards the ground and the Cartesian distance to the closest airports is computed. If there is no airport within 10 kilometers, the value is set to 'NULL'. Else, the closest identified airport is listed as the destination airport. The procedure applies in reverse

for the origin airport candidates.

We note that this approach is necessarily an extrapolation and airports may in some cases be wrongly identified if the contact is lost before the ground, in particular where several airports are close by.

### 3.3    Data Cleaning

To make the data accessible and meet the requirements, complex pre-processing is needed to abstract from most system as-

pects, reduce the data volume, and to eliminate the need to understand all system aspects in order to use the data. Moreover, the information quality needs to be assessed and indicated, allowing researchers to choose subsets that match their own require-

ments. Therefore, we performed the following processing steps to prepare the unstructured OpenSky Network data and create a well-defined dataset for scientific analysis.

### 3.3.1 Decoding

Decoding ADS-B correctly is a complex task. Although libraries and tutorials such as Sun et al. (2019) exist, it remains a tedious task that requires a deep understanding of the underlying link layer technology Mode S. Moreover, the sheer volume of data collected by OpenSky (about 120 GB of raw data per hour) makes this process challenging and resource-intensive. Therefore, we relieve researchers from this burden by providing readily decoded information such as position in WGS84 coordinates, altitude information in meters, and the unique aircraft identifier as a 24 bit hexadecimal number.

### 3.3.2 Timestamps

Timestamps are provided in different resolutions and units, depending on the receiving sensor type. For purposes of this dataset, we use the time when the messages where received at the server, with a second precision, which we deem more than sufficient for the macro use cases intended.

### 3.3.3 Deduplication

OpenSky's raw data is merely a long list of single measurements by single sensors. However, as most localization algorithms rely on signals being received by multiple receivers, we grouped multiple receptions belonging to the same transmission based on their continuous timestamp and signal payload. This process is called deduplication. Note that although most position reports are unique, a small number of falsely grouped measurements remains as noise in the data.

### 3.3.4 Quality Assurance

Crowdsourcing creates potential issues regarding the quality and integrity of location and timing information of certain aircraft and sensors. Such issues can range from faults in the transmission chain (i.e., aircraft transponder, ground station) to malicious injection of falsified aircraft data. To allow researchers to ignore these effects while still preserving them as a potential subject of research, OpenSky offers integrity checks to verify and judge the data correctness (see Schäfer et al. (2018)). We also note that the abstracted nature of this dataset makes it more robust to any issues in the first place as low-quality data will be averaged 120 out over time by the many involved receivers.

## 3.4 Data Enrichment

We use the OpenSky aircraft database to add aircraft types to our flight data, and access publicly available open application programming interfaces (API) to match the commercial flight identifier, where available.



The integration of aircraft types enables additional analysis such as gauging the popularity of different types and manu-
facturers across time, regions and use cases. Aircraft type designators follow the International Civil Aviation Organisation's
(ICAO) convention (International Civil Aviation Union (2020)).

The OpenSky aircraft database was created in 2017 as an additional crowdsourcing project. It joins different data sources,
official and unofficial ones. The official sources include the registration information from the flight authorities in the US, UK,
Ireland and Switzerland, which is downloaded and incorporated daily. Besides these, it relies on enthusiast knowledge based
on live observations and third-party sources. These are integrated opportunistically; the database is editable by any registered
user of the OpenSky Network. A static snapshot of the database from June 2020 is provided with this record, regular updates
are made available at https://opensky-network.org/datasets/metadata/.

## 4 Data Records

Overall, the archive includes 41,900,660 flights, from 160,737 aircraft, which were seen to frequent 13,934 airports in 127
countries. One file per month is provided in the comma-separated values (CSV) format. Table 1 provides a breakdown of the
included CSV files and their contents, broken down into size, number of flights, number of unique aircraft, unique origins and
destinations. Note the significant reduction in size and flights since the beginning of pandemic measures in March 2020.

We describe the columns of the dataset in the following:

1. `callsign`: The identifier of the flight used for display on the radar screens of air traffic controllers or communication
   over voice. For commercial flights, the first three letters are typically reserved for an airline, e.g. AFR for Air France,
   DLH for Lufthansa. This is then typically followed by four digits. For non-airline flights this can often be chosen freely
   or depending on the customs of the airspace of a country. It is broadcast by the airplane itself. For anonymity reasons,
   the callsign is only provided for verified commercial airline flights.

2. `number`: The commercial number of the flight, if available through OpenSky. These flight numbers are typically used
   by the airlines for booking references or departure boards at airports.

3. `aircraft_uid`: A unique aircraft identification number randomly generated based on the transponder identification
   number that is globally unique and specific to an aircraft (rather than a flight). Changes occur only if an aircraft changes
   ownership, with exceptions for military aircraft, which may in some countries be able to change their identifier at will.

4. `typecode`: The aircraft model type if available through the aircraft database.

5. `origin`: A four letter code for the origin airport of the flight, if the trajectory could be matched successfully.

6. `destination`: A four letter code for the destination airport of the flight, if the trajectory could be matched successfully.

7. `firstseen`: The UTC timestamp of the first message received by the OpenSky Network.



**Table 1.** Overview of the dataset files and content metadata.

| Filename | Month | Size | Aircraft | Flights |
|---|---|---|---|---|
| flightlist_20190101_20190131 | Jan 2019 | 175.5MB | 68,876 | 2,145,469 |
| flightlist_20190201_20190228 | Feb 2019 | 164.0MB | 68,798 | 2,005,958 |
| flightlist_20190301_20190331 | Mar 2019 | 186.5MB | 74,362 | 2,283,154 |
| flightlist_20190401_20190430 | Apr 2019 | 194.6MB | 76,298 | 2,375,102 |
| flightlist_20190501_20190531 | May 2019 | 208.2MB | 79,547 | 2,539,167 |
| flightlist_20190601_20190630 | Jun 2019 | 218.3MB | 82,879 | 2,660,901 |
| flightlist_20190701_20190731 | Jul 2019 | 238.3MB | 86,385 | 2,898,415 |
| flightlist_20190801_20190831 | Aug 2019 | 246.0MB | 89,776 | 2,990,061 |
| flightlist_20190901_20190930 | Sep 2019 | 224.1MB | 89,963 | 2,721,743 |
| flightlist_20191001_20191031 | Oct 2019 | 242.3MB | 92,449 | 2,946,779 |
| flightlist_20191101_20191130 | Nov 2019 | 223.5MB | 92,003 | 2,721,437 |
| flightlist_20191201_20191231 | Dec 2019 | 222.1MB | 92,253 | 2,701,295 |
| flightlist_20200101_20200131 | Jan 2020 | 225.4MB | 90,821 | 2,734,791 |
| flightlist_20200201_20200229 | Feb 2020 | 218.0MB | 97,931 | 2,648,835 |
| flightlist_20200301_20200331 | Mar 2020 | 177.2MB | 94,631 | 2,152,157 |
| flightlist_20200401_20200430 | Apr 2020 | 68.3MB | 74,257 | 842,905 |
| flightlist_20200501_20200531 | May 2020 | 87.8MB | 89,721 | 1,088,267 |
| flightlist_20200601_20200630 | Jun 2020 | 116.9MB | 98,747 | 1,444,224 |
| **All files** | **17 months** | **3.4 GB** | **160,737** | **41,900,660** |

8. `lastseen`: The UTC timestamp of the last message received by the OpenSky Network.

9. `day`: The UTC day of the last message received by the OpenSky Network.

10. `latitude_1, longitude_1, altitude_1` The position of the aircraft at the `firstseen` timestamps. The altitude is a barometric measurement based on a standard pressure of 1013 HPa.

11. `latitude_2, longitude_2, altitude_2` The position of the aircraft at the `lastseen` timestamps. The altitude is a barometric measurement based on a standard pressure of 1013 HPa.





## 5 Technical Validation

In the following, we provide some statistics showing that our flights dataset reflects the air traffic reality as different time series showing the effect of the COVID-19 pandemic at different airports and for different airlines.

**Table 2.** Flight distribution in data set January 2020.

| Manufacturer | Model | Typecode | Flights |
|---|---|---|---|
| Airbus | A320-neo | A20N | 77,018 |
| Airbus | A321-neo | A21N | 18,411 |
| Airbus | A319 | A319 | 116,261 |
| Airbus | A-320 | A320 | 365,901 |
| Airbus | A-321 | A321 | 128,332 |
| Airbus | A330-200 | A332 | 20,958 |
| Airbus | A330-300 | A333 | 39,618 |
| ATR | ATR-72-600 | AT76 | 35,788 |
| Boeing | 737-700 | B737 | 109,362 |
| Boeing | 737-800 | B738 | 378,424 |
| Boeing | 737-900 | B739 | 44,131 |
| Boeing | 757-200 | B752 | 29,318 |
| Boeing | 767-300 | B763 | 28,916 |
| Boeing | 777-200 | B772 | 17,147 |
| Boeing | 777-300/ER | B77W | 36,925 |
| Boeing | 787-9 Dreamliner | B789 | 19,085 |
| Bombardier | CRJ200 | CRJ2 | 46,930 |
| Bombardier | CRJ700 | CRJ7 | 29,829 |
| Bombardier | CRJ900 | CRJ9 | 49,293 |
| De Havilland | DHC-8-400 | DH8D | 34,487 |
| Embraer | ERJ 145 | E145 | 27,150 |
| Embraer | ERJ 190 | E190 | 27,871 |
| Embraer | ERJ 175 (long wing) | E75L | 43,891 |
| Embraer | ERJ 175 (short wing) | E75S | 21,251 |
| Pilatus | Eagle | PC12 | 16,663 |

Table 2 shows the distribution of the top 25 aircraft types in the flight dataset over one month (January 2020). Overall, the top models are dominated by the four largest commercial aircraft manufacturers: Boeing with 8 different types accounting

for 663,308 flights; Airbus with 7 models and 766,499 flights; Embraer with 4 models (120,163 flights) and Bombardier (3 models, 126,052 flights). The 737-800 is the single most popular aircraft with 378,424 flights in January 2020 alone.





**Table 3.** Top 20 airports based on recorded flight destinations in January 2020.

| Country | City | ICAO Code | Landings |
|---|---|---|---|
| United States | Atlanta | KATL | 35,770 |
| United States | Chicago | KORD | 34,480 |
| United States | Dallas–Fort Worth | KDFW | 27,534 |
| United States | Los Angeles | KLAX | 23,659 |
| United States | Las Vegas | KLAS | 21,195 |
| United States | Phoenix | KPHX | 19,219 |
| United States | New York Newark | KEWR | 18,962 |
| United Kingdom | London Heathrow | EGLL | 18,340 |
| United States | San Francisco | KSFO | 17,824 |
| United States | New York JFK | KJFK | 17,653 |
| United States | Houston | KIAH | 17,626 |
| India | New Delhi | VIDP | 17,498 |
| United States | Miami | KMIA | 17,154 |
| France | Paris CDG | LFPG | 16,889 |
| Malaysia | Kuala Lumpur | WMKK | 16,726 |
| United States | Seattle | KSEA | 16,670 |
| United Arab Emirates | Dubai | OMDB | 16,049 |
| Canada | Toronto | CYYZ | 15,972 |
| United States | Boston | KBOS | 15,927 |
| Germany | Frankfurt am Main | EDDF | 15,904 |

Table 3 shows the distribution of the top 20 airports types in the flight dataset in January 2020 (based on recorded flight destinations). Reflecting both global air traffic realities and OpenSky's coverage focus, 13 of these airports are in the United States, including the 7 busiest with regards to landings. Several of the major hubs in Europe (Frankfurt, London Heathrow, Paris Charles de Gaulle) and Asia (Kuala Lumpur, Dubai and Delhi) make up the remaining six.

Figure 4 shows a time series of airport activity (as measured by departures) on four different regions based on data from 1 January to 30 April 2020. The impact of the pandemic (or rather the measures to contain it) can be seen clearly in all four. For example, the data shows:

    – a slow decrease from February in several East-Asian airports (even earlier in Hong Kong);

    – European airports decreasing sharply from early March onward;

    – America's air traffic started dropping later by about two weeks;

    – India stopping all air traffic sharply by mid-March (VABB, VIDP).





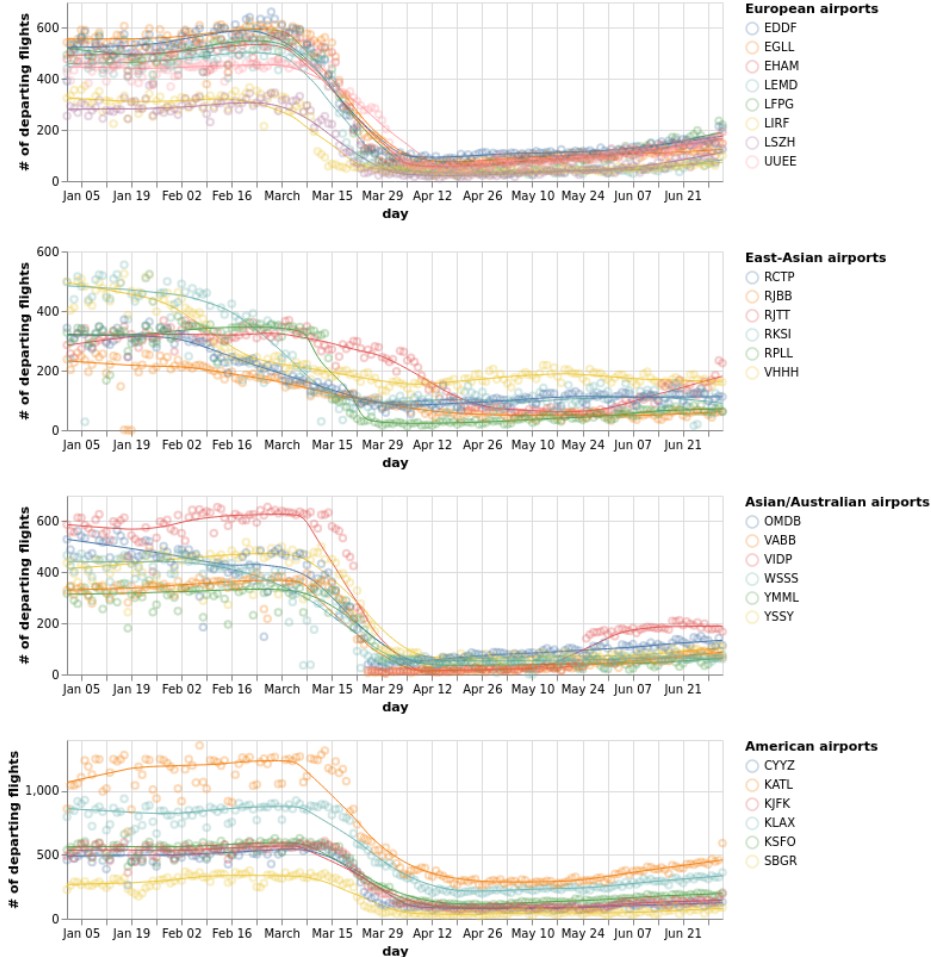

**Figure 4.** Comparison of flight numbers at various airports as seen by the OpenSky Network during 2020.

In a similar fashion, Figure 5 shows COVID-19's normalized impact on different airlines across the globe. Among noticeable trends, we can identify:

– sharply decreasing patterns for all regular airlines in March, with stronger effects for European airlines compared to American and Asian airlines;

    – almost all low-cost airlines practically stopped all business activities (with the exception of the Japanese Peach airlines);

    – a very slow recovery for most airlines and regions beginning in May and June, with some rebounding more strongly, for example Air New Zealand (ANZ);

– cargo airlines show no negative impact of the crisis, some may even find a slight upwards trend.



**Figure 5.** Comparison of flight numbers of various airlines as seen by the OpenSky Network during 2020. Flights are grouped by geographic regions for legacy carriers. Cargo and low-cost airlines are shown separately.



## 6 Usage Notes

This dataset may differ from other data sources due to limitations of ADS-B data. On the other hand, there are advantages as it reflects all aircraft types rather than only commercial airlines.

It is important to note that ADS-B equipage has been increasing over time as existing aircraft have been retrofitted and older aircraft have been replaced. This effectively means that the number of tracked aircraft in the dataset has been slowly increasing pre-pandemic, reflecting the reality of a dynamic global aviation industry.

Further, there are differences in ADS-B equipage across countries' airspaces (depending on their regulatory approach) as well as potentially between aircraft types. For example, small personal aircraft flying locally and below 18,000 feet are often not required to use ADS-B. Similarly, military aircraft may have exceptions for operational reasons. It is not possible to track and reflect these highly dynamic developments in a static dataset, however, this should be kept in mind for comparative analysis purposes.

Finally, as a recommendation for data handling and visualization, Figures 4 and 5 have been created with the open-source Python package *traffic* (Olive (2019)), which offers dedicated methods for air traffic data and interfaces with OpenSky and other data sources.

## 7 Conclusions

Air traffic and flight data is needed for effective research in many areas of Earth Systems Science and related fields. We presented an openly accessible, specifically crafted dataset based on crowdsourced data obtained through the OpenSky Network and validated it successfully. From January 2019 to July 2020, the archive includes 41,900,660 flights, from 160,737 aircraft, which were seen to frequent 13,934 airports in 127 countries. As it is updated monthly, this dataset will be growing significantly and provide deeper insights into flight behaviour before, during, and after the COVID-19 pandemic.

## 8 Code and data availability

The dataset is available under the CC-BY license at Zenodo (see Olive et al. (2020), DOI: https://doi.org/10.5281/zenodo.3928564).

The code to generate and process the data is available in different components. The popular dump1090 package, used as the basis to receive a large majority of crowdsourced information (ca. 80% in OpenSky), is available at Foster (2017). Other receiver software may include proprietary and closed source software such as Radarcape and SBS-3.

The OpenSky decoder is available in OpenSky's GitHub repository at https://github.com/openskynetwork/java-adsb.

Code concerning data cleaning and processing is documented at https://traffic-viz.github.io/scenarios/covid19.html.



*Author contributions.* M.St. manuscript writing and data collection; X.O. data preparation, data cleaning, data visualization; J.L. route data
collection; M.Sc. and V.L. OpenSky Network infrastructure. All authors manuscript review.

*Competing interests.* The authors declare that they have no competing interests.

*Acknowledgements.* We are thankful to all members and supporters of the OpenSky Network foundation for making this data collection
effort possible.





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
