# Peer review of "Crowdsourced Air Traffic Data from the OpenSky Network 2019–20"

_Earth System Science Data, 2020_

## Referee Comment (RC1) · Simon Proud (Referee) · 2 Sep 2020

Overall this is a very useful dataset, and the manuscript describes it well. I have no major recommendations for changes but there are a few minor issues and also some questions that I have:

Abstract line 2: Change 'airplanes' to 'aircraft', as Opensky also sees helicopters etc.

Line 5: Delete 'further' from 'surged further'.

Line 9: I note that the text says the dataset ends on Jul 1st 2020. Will this be extended? I know the paper describes the data up to that date, but if the actual dataset will be continued beyond this then it'd be worthwhile to note in the text.

[Figure]

Line 17: Change 'article' to 'particulate'.

Line 31: I think a typo: 'xte'.

Figure 1: You might need to explain what the 'stopped anonymous feeding' note on this figure is, as readers unfamiliar with opensky might get confused here.

Lines 74+75: You say "many well-covered regions" and "many countries" in the same sentence. One of these can be deleted.

Lines 83+84: The description of the method to prevent counting multiple flights is unclear. This sentences needs rewording. From what I can gather, the speed and distance are extrapolated, and if a new position report is received then its location is sanity-checked against this extrapolated speed/distance/time. But please reword, and also include a bit more detail (how close does the extrapolated time have to be, for example?)

Line 91 + 92: Could callsign be used to verify this? I presume it'd mean having to call in external data, but could be a useful check of how accurate the estimated departure/arrival airport is. Some kind of accuracy assessment would be very good here, as it is currently missing.

Line 105: What happens if multiple copies of the same message are received but at different times? Which timestamp is used?

Line 107: Change 'a second' to 'one second'.

Line 152 + 153: Do the first seen and last seen times include on ground reports? I'm thinking about aircraft that sit on the ground with the transponder on, or get stuck in queues for the runway.

Line 156 + 157: Likewise with altitude, is the altitude reported the first one above the groud, or the on-ground altitude? I have noticed from the actual ADS-B data in opensky that on ground altitudes are often incorrect (10000+ft is not uncommon) so how is this

handled?

Line 155: Could an 'average altitude' for the flight be added to the data? This would be really useful for a large number of environmental researchers, for example. Figures 4 + 5: I find these plots quite hard to read (they're much nicer in the 'traffic' html than in a PDF document!) Could you please re-work these figures to make them more suitable for the print / pdf copy? Maybe remove the circles and just keep the lines?

Line 181: Change "Almost all" to "Most".

---

## Author Comment (AC1) · 9 Oct 2020

First of all, we all appreciate the very thoughtful review of Dr. Proud. We are very happy to take the suggestions on board. We will fix all the unnecessary typos and text issues, of course, thanks for pointing them out so thoroughly. In the following, we will respond to the remaining comments:

1. Yes, the dataset is updated monthly and now available until the end of August 2020. We will post the update for September in the coming days. Originally, we planned this as a special service during the pandemic, which will clearly still go on for the foreseeable future. Depending on demands and time constraints we may choose other than monthly update cycles at some point in the future.

[Figure]

2. We will add a comment about the discontinued anonymous feeding feature (and refer the very interested reader to [1], where it is discussed in detail).

3. The accuracy verification of the airport predictions using external data is an excellent idea! We will look into what kind of data we can obtain to run such an assessment for future updates.

4. The first timestamp is used in case the message is received multiple times (which is indeed typically the case).

5. The flight separation (incl. first/last seen times and altitude) is done on airborne reports. Depending on the use case other design choices may be preferable (where poeple are interested in taxiing/block times) but we did not want to add too many fields to the dataset at this point. If there's the demand for it, we are happy to investigate it for future updates. Similarly, we will look into the added processing requirement for the average altitude.

6. Thanks for pointing out the readability issues in Figures 4 + 5, which are indeed web-optimised. We will refactor them for the revised/finalised version.

[1] Schäfer, M., Strohmeier, M., Smith, M., Fuchs, M., Lenders, V., and Martinovic, I.: OpenSky report 2018: assessing the integrity of crowd-sourced mode S and ADS-B data, in: 2018 IEEE/AIAA 37th Digital Avionics Systems Conference (DASC), pp. 1–9, IEEE, 2018.

---

## Short Comment (SC1) · 30 Nov 2020

The paper very clearly describes a high quality OpenSky dataset that provides an overview of the global flights during the year 2019 and 2020. The flight information is provided in the format of origin-destination pair. Even without detailed trajectory information, this is a valuable data source for macro-level air transportation related studies.

My only comment is about these flights that are out of OpenSky's coverage:

It might be useful for users to know what percentage of global flights is included in this dataset. Specifically, what is the total miles of flights in the dataset compare to the global flight statistics. I believe this information, even a rough estimate, can be a great

help for many follow-on studies that make use of this dataset.

---

## Referee Comment (RC2) · Anonymous Referee #2 · 1 Dec 2020

Data cleanly available as monthly compressed files from Zenodo. Very useful metadata and guidance on Zenodo landing page.

Decoding step particularly useful!

Very good description, no comments from this reviewer. A few rare typos, probably corrected during proof-reading.

Biggest concern = file size. Unpacking the .gz file results in large (0.5 GB) .csv files, large for some spreadsheet software on some computers. This reviewer never succeeded to load May 2019 data into a Google spreadsheet, not from computer file system nor from Google drive, despite working on a reasonably fast home network. Evidently Google spreadsheets operate with a file size limits of 40 MB and 400k cells. For older versions of Microsoft Excel on older Windows PC operating systems, similar limits will apply? Many users will confront these barriers? Once opened into Mac Numbers (similar to Excel), files produce a clean useful well-documented output albeit with absurd precision of many numbers in many cells. Author and editors need to fix the file size problem for purposes of this document and for future users. Not a good situation if a reviewer can not use usual tools to access data.

Once they resolve the file size barriers, this should represent a nice product. When published, ESSD should join with authors to make public announcement?

---

## Author Response (AR1)

**Dear Reviewers and Editor,**

**We thank you very much for your service and the extensive reviews. We address the responses in detail below.**

**Best wishes,**

**Martin Strohmeier, Xavier Olive, Jannis Lübbe, Matthias Schäfer, Vincent Lenders**

**Reviewer 1:**

RC1: Abstract line 2: Change 'airplanes' to 'aircraft', as Opensky also sees helicopters etc.

**Change: Fixed as suggested.**

RC1: Line 5: Delete 'further' from 'surged further'.

**Change: Fixed as suggested.**

RC1: Line 9: I note that the text says the dataset ends on Jul 1st 2020. Will this be extended?I know the paper describes the data up to that date, but if the actual dataset will be continued beyond this then it'd be worthwhile to note in the text.

**Response: Yes, the dataset is updated monthly and now available until the end of November 2020. We will post the update for December in the coming days. Originally, we planned this as a special service during the pandemic, which will clearly still go on for the foreseeable future, in particular as effects on aviation go. We have automated this update process as far as is possible with the current features.**

**Change: We have updated the last two sentences of the abstract to: "It spans all flights seen by the network's more than 3500 members between 1 January 2019 and 1 July 2020. The archive is being updated every month and for the first 18 months includes 41,900,660 flights, from 160,737 aircraft, which were seen to frequent 13,934 airports in 127 countries."**

RC1: Line 17: Change 'article' to 'particulate'.

**Change: Fixed as suggested.**

RC1: Line 31: I think a typo: 'xte'.

**Change: Fixed as suggested.**

RC1: Figure 1: You might need to explain what the 'stopped anonymous feeding' note on this figure is, as readers unfamiliar with opensky might get confused here.

**Response: Thanks for the suggestion. We have removed this from the picture as it is irrelevant to the dataset and would require explaining much additional history and background on technology. We have however referenced the relevant work about OpenSky's history here.**

**Change: Removed suggested text from Figure 1. Added reference to [1].**

RC1: Lines 74+75: You say "many well-covered regions" and "many countries" in the same sentence. One of these can be deleted.

**Change: Fixed as suggested.**

RC1: Lines 83+84: The description of the method to prevent counting multiple flights is unclear. This sentences needs rewording. From what I can gather, the speed and dis-tance are extrapolated, and if a new position report is received then its location is sanity-checked against this extrapolated speed/distance/time. But please reword, and also include a bit more detail (how close does the extrapolated time have to be, for example?)

**Response: The interpretation is correct. We extrapolate the part of the flight that our network cannot see based on the speed and distance of an aircraft when it leaves our coverage, say, from Europe over the Atlantic Ocean. When the same aircraft is seen again within sensible margins of error entering our coverage in the US, it is treated as the same flight.**

**Change: We have rewritten the respective Section 3.2 for increased clarity.**

RC1: Line 91 + 92: Could callsign be used to verify this? I presume it'd mean having to call in external data, but could be a useful check of how accurate the estimated depar-ture/arrival airport is. Some kind of accuracy assessment would be very good here, asit is currently missing.

**Response: The accuracy verification of the airport predictions using external data is an excellent idea. It is however difficult, as if such data were freely available, our dataset would likely not even be needed in the first place! However, we are working with Eurocontrol to be able to verify at least the European portion of the data in the future and will integrate this into the dataset / description when this project is finalised.**

**Change: No change is possible yet as we need to acquire reliable data from external sources but we are working to integrate such an assessment for future updates.**

RC1: Line 105: What happens if multiple copies of the same message are received but at different times? Which timestamp is used?

**Response: The first timestamp is used in case the message is received multiple times (which is indeed typically the case).**

**Change: Added a sentence to this effect to Section 3.3.2.**

RC1: Line 107: Change 'a second' to 'one second'.

**Change: Fixed as suggested.**

RC1: Line 152 + 153: Do the first seen and last seen times include on ground reports? I'm thinking about aircraft that sit on the ground with the transponder on, or get stuck in queues for the runway.

Line 156 + 157: Likewise with altitude, is the altitude reported the first one above the ground, or the on-ground altitude? I have noticed from the actual ADS-B data in openskythat on ground altitudes are often incorrect (10000+ft is not uncommon) so how is this handled?

Line 155: Could an 'average altitude' for the flight be added to the data? This would be really useful for a large number of environmental researchers, for example.

**Response: The flight separation (incl. first/last seen times and altitude) is done on airborne reports. Indeed ground reports are often unreliable due to transponder issues, which would need to be worked/filtered out.**

**Depending on the use case other design choices may be preferable (where people are interested in taxiing/block times) but we did not want to add too many fields to the dataset at this point (reviewer 2 already complains about the size!). If there's the demand for it, we are happy to investigate it for future updates, however. Similarly, we will look into the added processing requirement for adding the average altitude in future updates, which requires to parse each flight fully rather than just the beginning/end.**

**Change: We have added the word "airborne" to the explanations of "firstseen" , "lastseen" and "day" in the "Data Records" section.**

RC1: Figures 4 + 5: I find these plots quite hard to read (they're much nicer in the 'traffic' html than in a PDF document!) Could you please re-work these figures to make them more suitable for the print / pdf copy? Maybe remove the circles and just keep the lines?

**Response: Thanks for pointing out the readability issues in Figures 4 + 5, which are indeed web-optimised.**

**Change: We have adapted Figures 4 and 5.**

RC1: Line 181: Change "Almost all" to "Most"

**Change: Fixed as suggested.**

**Reviewer 2:**

RC2: Biggest concern = file size. Unpacking the .gz file results in large (0.5 GB) .csv files, large for some spreadsheet software on some computers. This reviewer never succeeded to load May 2019 data into a Google spreadsheet, not from computer file system nor from Google drive, despite working on a reasonably fast home network. Evidently Google spreadsheets operate with a file size limits of 40 MB and 400k cells. For older versions of Microsoft Excel on older Windows PC operating systems, similar limits will apply? Many users will confront these barriers? Once opened into Mac Numbers (similar to Excel), files produce a clean useful well-documented output albeit with absurd precision of many numbers in many cells. Author and editors need to fix the file size problem for purposes of this document and for future users. Not a good situation if a reviewer can not use usual tools to access data.

**Response: We appreciate this perspective that we had not considered before. Our dataset is still relatively small but overall we would put it into the category of 'big data'. In particular it is an abstraction of two petabytes of data, covering the whole OpenSky dataset within a small fraction (0.00018%!). While it would be possible to split it into, say, weekly files, we believe this would make processing rather more complicated for most purposes (e.g., clear end of year breaks). Instead, we provide tools, tutorials and usage examples on several supporting websites, e.g. [1] and [3].**

**Change: Creation of dynamic websites and tool/handling discussions, mentioned in the paper in Section 6: "Further usage notes and tool recommendations are regularly added on the OpenSky Website (https://opensky-network.org/community/blog/item/6-opensky-covid-19-flight-dataset). "**

**Reviewer 3 (Short Comment):**

RC2: It might be useful for users to know what percentage of global flights is included in this dataset. Specifically, what is the total miles of flights in the dataset compare to the global flight statistics. I believe this information, even a rough estimate, can be a great help for many follow-on studies that make use of this dataset.

**Response: Thanks for the suggestion. Exact numbers for all types of flights are difficult to come by, due to the nature of this global system. The best estimate we could find comes from the commercial flight tracker website FlightRadar24 [4]. It mentions tracking 68,948,849 total flights in 2019, sadly not breaking it down into how many flights were tracked via ADS-B in order to be able to directly compare 
[revised manuscript text omitted]